# LANGUAGE GUIDED SKILL DISCOVERY

**Seungeun Rho**
Georgia Institute of Technology
srho31@gatech.edu

**Laura Smith**
University of California, Berkley
smithlaura@berkeley.edu

**Tianyu Li**
Georgia Institute of Technology
tli471@gatech.edu

**Sergey Levine**
University of California, Berkley
svlevine@eecs.berkeley.edu

**Xue Bin Peng**
Simon Fraser University
xbpeng@sfu.ca

**Sehoon Ha**
Georgia Institute of Technology
sehoonha@gatech.edu

## ABSTRACT

Skill discovery methods enable agents to learn diverse emergent behaviors without explicit rewards. To make learned skills useful for downstream tasks, obtaining a semantically diverse repertoire of skills is crucial. While some approaches use discriminators to acquire distinguishable skills and others focus on increasing state coverage, the direct pursuit of 'semantic diversity' in skills remains underexplored. We hypothesize that leveraging the semantic knowledge of large language models (LLM) can lead us to improve semantic diversity of resulting behaviors. In this sense, we introduce Language Guided Skill Discovery (LGSD), a skill discovery framework that aims to directly maximize the semantic diversity between skills. LGSD takes user prompts as input and outputs a set of semantically distinctive skills. The prompts serve as a means to constrain the search space into a semantically desired subspace, and the generated LLM outputs guide the agent to visit semantically diverse states within the subspace. We demonstrate that LGSD enables legged robots to visit different user-intended areas on a plane by simply changing the prompt. Furthermore, we show that language guidance aids in discovering more diverse skills compared to five existing skill discovery methods in robot-arm manipulation environments. Lastly, LGSD provides a simple way of utilizing learned skills via natural language.

## 1 INTRODUCTION

One of the key capabilities of intelligent agents is to autonomously learn useful skills applicable to downstream tasks without task-specific objectives. Consider how human infants acquire manipulation skills through simple play with toys, which can later be applied to a broader set of tasks, such as using a spoon or holding a bottle. Our research aims to emulate this process of skill learning. Prior works in unsupervised skill discovery suggest that maximizing diversity of behaviors can be a way to develop such skills (Gregor et al., 2016; Eysenbach et al., 2018; Kwon, 2020). These approaches operate under the assumption that acquiring a wide range of diverse behaviors may lead to the development of useful skills for downstream tasks. Building on this premise, studies like Sharma et al. (2019); Hansen et al. (2019); Liu & Abbeel (2021a); Laskin et al. (2022) have associated behaviors with random vectors by maximizing Mutual Information between them to acquire diverse skills. Additionally, there are other approaches aimed at increasing coverage of the state space (Liu & Abbeel, 2021b; Yarats et al., 2021; Park et al., 2021), or exploration-based strategies that leverage errors in predictive models to enhance learning (Burda et al., 2018; Pathak et al., 2019; Park et al., 2023a).

However, we contend that these measures are proxies for what we term "**semantic diversity**" and they do not necessarily reflect the semantic diversity of a repertoire of skills. For instance, in high

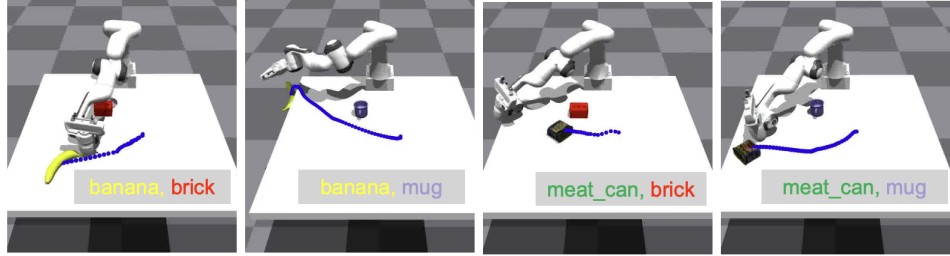

Figure 1: We proposed LGSD which can discover a semantically distinctive set of skills. We showcase four sample skills acquired from a single training run. Our approach successfully learned skills that manipulate only 'edible' objects (banana and meat_can) from a total of four objects.

degrees of freedom (DOF) systems, such as robot-arm manipulation tasks, unstructured movements of each DOF might exhibit high diversity in terms of state coverage or distinctiveness from a neural discriminator, but nonetheless lack meaningful semantic diversity. To more accurately measure semantic diversity, we propose the use of Large Language Models (LLM) as they are well-suited for understanding and assessing the semantic significance of different behaviors in a way that better aligns with human judgment.

In this work, we present **Language Guided Skill Discovery (LGSD)**, a skill discovery algorithm that utilizes the guidance of LLMs to learn semantically diverse skills. LGSD aims to fulfill three desiderata for skill discovery. (i) Firstly, we want to discover skills that have diverse semantic meanings. We use LLMs to generate descriptions for each of agent states. Based on these descriptions, we measure the semantic difference between states using *language-distance*, and train skills to maximize this distance. Because LLMs are equipped with a semantic understanding of the world, this provides a way to maximizing semantic differences between skills. Fig. 1 illustrates skills learned from LGSD, showcasing the semantic diversity of resultant behaviors. (ii) Secondly, we seek to restrict the search space for skills into a desired semantic subspace. Intuitively, restricting the skill space is akin to enforcing an infant to grab toys only with their hands, and not with their mouth or feet. We implement this desideratum by utilizing language prompts. Humans provide prompts to LLMs, and the generated descriptions are then constrained by these prompts. This will allow our method to ignore differences outside the intended semantic space. (iii) Lastly, we aim for the learned skills to be easy to use. When the skill space is continuous, merely selecting the best skills for target tasks becomes non-trivial. LGSD supports a zero-shot language instruction-following capability by training separate network $\psi$ for skill inference. Humans can provide a natural language description of the desired state, and then LGSD can infer which skill should be used to reach that goal state.

The primary contributions of this work are as follows. 1) We propose a skill discovery framework that enables the discovery of semantically diverse skills by utilizing language guidance from LLMs. To the best of our knowledge, this is the first work to incorporate semantic diversity into skill discovery. 2) We present a method of constraining the skill search space to a user-defined semantic space using language prompts. This allows users to control the resulting skills by simply specifying different prompts. We emphasize that, prior to training, existing methods only offer limited control over the skills to be learned. 3) We provide a theoretical proof of how *language-distance*, a proxy for semantic distance, can be employed as a valid *pseudometric*. 4) We propose a method that enables an agent to reach a goal state specified by natural language, facilitating the convenient use of learned skills. 5) We demonstrate that our method can train semantically diverse skills. Through extensive experiments, we show that LGSD outperforms five different skill discovery baseline methods on both locomotion and manipulation tasks, in terms of both diversity and sample efficiency.

## 2 RELATED WORK

### 2.1 UNSUPERVISED SKILL DISCOVERY

**Mutual information based approach**  Common approaches to associate skills with corresponding behaviors are based on maximizing the mutual information (MI) between states $S$ and skills $Z$, or $\mathcal{I}(S; Z)$ (Gregor et al., 2016; Eysenbach et al., 2018; Sharma et al., 2019; Hansen et al., 2019; Liu & Abbeel, 2021a; Laskin et al., 2022). MI can be viewed as a dependency measure that quantifies the association between two random variables. To maximize this quantity, a popular early formulation

utilizes the decomposition $\mathcal{I}(S; Z) = \mathcal{H}(Z) - \mathcal{H}(Z|S)$, where $\mathcal{H}[\cdot]$ refers to Shannon entropy. Since directly computing $\mathcal{H}(Z|S)$ is intractable, Eysenbach et al. (2018) introduce a variational posterior $q_\phi(z|s)$ to optimize the lower bound. Similarly, CIC (Laskin et al., 2022) maximizes $\mathcal{I}((s, s'); Z)$, promoting distinct state transitions across different skills. However, a significant limitation of using MI as the objective arises from the use of Kullback–Leibler (KL) divergence to define MI:

$$\mathcal{I}(S; Z) \overset{\text{def}}{=} D_{KL}(P(S, Z) \| P(S)P(Z)).$$

Maximization of MI using KL divergence often results in skills with less distinctive behaviors. This is because KL divergence is fully maximized when two densities have no overlap, posing a problem as there is no further incentive to distinguish the densities beyond this point. This is problematic because having no overlap between two densities does not necessarily mean two skills are noticeably distinctive. In practice, this becomes a severe shortcoming of DIAYN (Eysenbach et al., 2018), where its objective can be fully optimized as long as the discriminator $q_\phi$ can distinguish them perfectly. Neural networks can easily distinguish minor numerical differences, so the skills learned with this objective often have only these minute differences.

**Distance maximization approach**    To overcome these shortcomings, recent skill discovery works (He et al., 2022; Park et al., 2021; 2023a;b) propose using the Wasserstein Dependency Measure (WDM, Ozair et al. (2019)):

$$\mathcal{I}_{\mathcal{W}}(S; Z) \overset{\text{def}}{=} \mathcal{W}(P(S, Z) \| P(S)P(Z))$$

WDM is defined using the Earth-Mover (EM) distance and retains the advantage of always providing an incentive to maximize the distance between distributions, even when they are already distinguishable.

An essential characteristic of $\mathcal{I}_{\mathcal{W}}$ is that it must be defined within a *metric* space. Therefore, the choice of *metric* to measure the difference between skills comes to the gist of the algorithm since it ultimately governs the resulting behavior of each skills. LSD (Park et al., 2021) employs the Euclidean distance in state space as its metric. Maximizing $\mathcal{I}_{\mathcal{W}}$ under this metric encourages agents to visit states that are are far apart in terms of Euclidean distances. Conversely, CSD (Park et al., 2023a) points out the problem of using Euclidean distance as the metric. They argue that maximizing Euclidean distance tends to focus on "easy" changes in the state space, such as moving the robot arm itself rather than the target object. Thus, CSD proposed to maximize "controllability-aware distance" instead. This metric measures the negative log likelihood of a transition dynamics $-\log p(s'|s)$. Using this metric means the algorithm favors visiting rare transitions. LGSD adopts a similar approach but differentiates itself by employing a language-based distance metric within the WDM framework, to maximize the semantic difference.

## 2.2    USING LANGUAGE AS GUIDANCE FOR BEHAVIOR LEARNING

Recent work has studied using semantic knowledge in large pre-trained models for control tasks in several ways. A popular approach is to use the common-sense reasoning capabilities of LLMs to produce high-level plans over known, existing skills through the interface of natural language (Sharma et al., 2021; Ahn et al., 2022; Singh et al., 2023; Huang et al., 2022; Ha et al., 2023) or generated programs (Liang et al., 2022; Zeng et al., 2022; Vemprala et al., 2024). In addition, LLMs have also been shown to be effective in directly producing low-level actions that can be executed by an agent (Wang et al., 2023; Kwon et al., 2023; Driess et al., 2023; Brohan et al., 2023). Our work does not use LLMs to generate *actions* at any level of abstraction but to provide a meaningful distance measure to guide learning towards discovering semantically meaningful skills.

Several works have used LLMs to guide agents to learn semantically meaningful behavior by using them to design reward functions for either a given desired task (Yu et al., 2023; Ma et al., 2023). This showcases LLMs' capacity to understand specific physical embodiments assorting it with semantic knowledge to reason about given tasks. We note that these works aim to solve a single specific task, whereas our method aims to learn a repertoire of diverse behaviors without explicit supervision.

The work most related to ours is Du et al. (2023), which employs an LLM to guide exploration. Their approach uses the LLM to suggest plausible set of goals for the agent, maximizing the cosine similarity between the language embeddings of the goals and the current state. While they focus

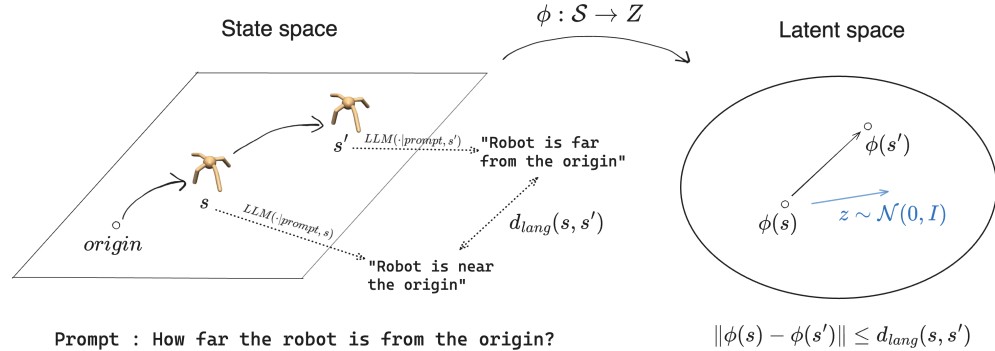

Figure 2: Overview of how LGSD works. Given a prompt, the LLM generates the description for each state. We then measure the difference between these descriptions and denote it as $d_{\text{lang}}$. Based on $d_{\text{lang}}$, we constrain the latent space by enforcing the 1-Lipschitz condition on $\phi$. Then the agent is encouraged to visit states that make the vector $\phi(s') - \phi(s)$ aligns well with a randomly sampled vector $z$ from an isotropic Gaussian prior. This makes the agent explore the latent space in diverse directions depending on the sampled $z$.

on using an LLM to reason with a low-dimensional, discrete set of high-level actions in complex environments like Crafter and Housekeep, our work focus on discovering meaningful behaviors in complex, continuous control spaces where actions (e.g. joint targets) are not as readily interpretable. Furthermore, their reward function, based on language similarity, lacks theoretical support as it is not a valid *metric*. In contrast, our method effectively handles the *language-distance* measure and provides a theoretical framework for its use. We elaborate on this in Section 4.3.

## 3 PRELIMINARIES

Unsupervised skill discovery can be formalized within the framework of a reward-free Markov Decision Process (MDP), denoted as $\mathcal{M} \equiv \{\mathcal{S}, \mathcal{A}, \mathcal{P}\}$. Here, $\mathcal{S}$ represents the state space, $\mathcal{A}$ denotes the action space, and $\mathcal{P} : \mathcal{S} \times \mathcal{A} \times \mathcal{S} \to [0, \infty)$ defines the transition dynamics. During training, a latent vector $z$ is sampled from a prior distribution $z \sim p(z)$ at the start of each episode and remains fixed throughout the episode. This vector informs the policy $\pi(a|s, z)$, guiding the generation of corresponding rollouts through $\pi$. The training procedure aims to foster a dependency of the latent vector $z$ on its resultant behavior, allowing it to function effectively as a *skill*. It is referred to as the *skill* because being conditioned on different skill $z$ aims to behave differently by encouraging agents to visit distinct distributions of states $s$, transitions $(s, s')$, or trajectories $\tau$. A common approach to training skills involves defining intrinsic rewards $r^{\text{int}} : \mathcal{S} \times \mathcal{A} \to \mathbb{R}$ and optimizing the accumulation of discounted intrinsic rewards with the following objective: $\mathcal{J} = \mathbb{E}_{\pi, \mathcal{P}} \left[ \sum_{t=0}^{\infty} \gamma^t r_t^{\text{int}} \right]$. Various off-the-shelf reinforcement learning (RL) algorithms can be employed to optimize $\pi$ to maximize this objective. Notably, this process does not utilize any task-related explicit rewards.

## 4 LANGUAGE GUIDED SKILL DISCOVERY

In this section, we begin by providing an overview of our algorithm, followed by in-depth explanations of the prompting, language distance measurement, and how we can utilize the acquired skills using language prompts.

### 4.1 ALGORITHM

Our goal is for an agent to learn a corpus of skills that visits semantically different states. Discovering semantically distinct states requires 1) a way of measuring the semantic difference between states, and 2) a means of maximizing this difference measure. To evaluate the semantic differences between different behaviors, we leverage the power of LLMs. We will consider two states as substantially distinct if the language descriptions produced by LLMs are semantically different. As

illustrated in Fig. 2, an LLM is queried to produce a natural language description for each state. This description links the state to its semantic meaning. We then measure the difference between these language descriptions. To measure the difference between two language descriptions, we leverage a pre-trained natural language embedding model, Sentence-Transformer (Reimers & Gurevych, 2019), to map the language descriptions into fixed-dimension real-valued vectors. This allows us to measure the difference between these vectors using cosine similarity. We refer to this difference measure as *language-distance* and denote it as $d_{\text{lang}}$.

Given a measure of the difference between language descriptions $d_{\text{lang}}$, which acts as a proxy for the semantic differences between states, the next step is to incentivize an agent to maximize this distance in order to discover diverse skills. We utilize existing distance maximization skill discovery algorithms, discussed in Section 2.1. Intuitively, we aim for agents to spread out from the origin in the latent space, which is constrained under $d_{\text{lang}}$. This constraint ensures that spreading out in the latent space leads to maximizing sum of $d_{\text{lang}}$ along the path in the state space. More specifically, we train a representation function $\phi : \mathcal{S} \rightarrow Z$ that maps state space $\mathcal{S}$ onto a latent space $Z$. Here, we enforce that $\phi$ is 1-Lipschitz continuous under $d_{\text{lang}}$, meaning that $\forall (s, s') \in \mathcal{S}$, $\|\phi(s') - \phi(s)\| \leq d_{\text{lang}}(s, s')$. Thus, the Euclidean distance between $s$ and $s'$ in latent space is always less than or equal to $d_{\text{lang}}(s, s')$. Hence, simply maximizing the Euclidean distance in the latent space results in maximizing $d_{\text{lang}}$ in the state space.

Spreading out in the latent space can be achieved by aligning the vector $\phi(s') - \phi(s)$ with a latent skill vector $z$ sampled from a Gaussian distribution $z \sim \mathcal{N}(0, I)$. Because the vector $z$ is sampled isotropically, aligning $\phi(s') - \phi(s)$ with $z$ fosters a diversity of behaviors depending on the sampled $z$. We use the reward $r(s, z, s') = (\phi(s') - \phi(s))^T z$ and maximize the cumulative reward to achieve this objective. Here, Park et al. (2023b) showed that maximizing the cumulative reward $r(s, z, s')$ with a 1-Lipschitz continuous $\phi$ is equivalent to maximizing the WDM between the terminal state $S_T$ and $Z$:

$$\arg\max_{\pi, \phi} \mathcal{I}_{\mathcal{W}}(S_T; Z) \approx \arg\max_{\pi, \phi} \mathbb{E}_{\pi, z \sim p(Z)} \left[ \sum_{t=0}^{T-1} (\phi(s_{t+1}) - \phi(s_t))^T z \right]$$
$$s.t. \ \forall (s, s') \in \mathcal{S}, \ \|\phi(s') - \phi(s)\| \leq d_{\text{lang}}(s, s') \tag{1}$$

We provide the proof in Appendix B. Therefore, the meaning of maximizing traveled language-distance is equivalent to maximizing the WDM between $s_T$ and $z$ under language-distance. Any off-the-shelf model-free RL algorithms can then be used to maximize the sum of rewards. We used PPO (Schulman et al., 2017) for as our primary RL algorithm because of its stable convergence. To ensure $\phi$ is 1-Lipschitz continuous under a distance metric $d$, we use dual gradient descent with a with a Lagrange multiplier $\lambda$ (Boyd & Vandenberghe, 2004).

## 4.2 Constraining the Search Space via Prompts

The potential space for skills is vast, yet only a narrow region possesses semantic meanings. We want to explore only within a subspace which has inherent semantic meanings. To achieve this desiderata, we use language prompts to constrain the entire search space into the desired semantic subspace.

Fig. 3 shows a manipulator environment that aims to move an object. In this environment, we can consider skills enabling object interactions as semantically meaningful while considering unstructured robot-arm movements not. Our language-based distance allows us to capture such delicate semantics of skills. For instance, we prompted an LLM to describe the distance $d_1$ between the robot arm and the object unless the object is moved already. Therefore, the scene description remains the same unless there is a change in the robot-object distance $d_1$. Because $d_{\text{lang}}$ becomes positive only when the description of a state is changed, our distance maximization algorithm encourages an agent to visit states that can change descriptions, which makes a robot arm to approach the object.

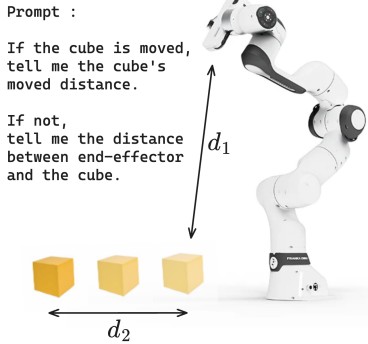

Figure 3: By prompting the LLM to generate different descriptions depending on the state, LGSD can adapt its focus during training.

Moreover, we highlight that we can induce the LLM to not only focus on a single semantic aspect of the scene but also to **adapt its focus** onto different semantic aspects throughout the training. For instance, when the robot arm finally reaches the cube, the LLM is then asked to describe the object's moved distance $d_2$. This additional prompt shifts the focus of the semantics to the object's traveled distance and any other changes in the states are ignored by the LLM. This leads to the robot arm pushing the object to its maximum extent. Note that existing skill discovery algorithms have limited capability to constrain its focus by feeding subset of observations (Eysenbach et al., 2018; Park et al., 2023a). In contrast, LGSD enables not only focusing on *semantic dimensions* but also adapting its focus across different semantics during training.

## 4.3 LANGUAGE DISTANCE AS A VALID METRIC

In Section 4.1, we stressed that $d_{\text{lang}}$ establishes a criterion for determining the distance between any two points in the latent space. Therefore, maintaining a coherent criterion is crucial for building the latent space. In this sense, $d_{\text{lang}}$ should be a valid distance *metric*. In this section, we present how $d_{\text{lang}}$ can be employed as a valid (pseudo-) *metric*, despite not satisfying the triangle inequality.

We begin with defining $d_{\text{lang}}$ formally. Let us denote a user prompt as $l_{\text{prompt}}$, and a rule-based state annotator as $g : \mathcal{S} \to l_{\text{state}}$. The annotator $g$ is a pre-defined function that translates state vector into corresponding natural language sentence without any addition or loss of information. Then, we denote the output description of an LLM for state $s$ as $l_{\text{desc}}(s) = LLM(\cdot|l_{\text{prompt}}, l_{\text{state}})$. We configure the LLM to be deterministic with zero temperature during generation, thus treating it as a function. Next, we employ a pre-trained language embedding model $f_{\text{embd}} : l_{\text{desc}} \to \mathbb{R}^N$. Using this setup, we can now define the language distance $d_{\text{lang}}$ as follows:

$$\forall (s, s') \in \mathcal{S}, \ d_{\text{lang}}(s, s') \stackrel{\text{def}}{=} 1 - \frac{f_{\text{embd}}(l_{\text{desc}}(s))^T \cdot f_{\text{embd}}(l_{\text{desc}}(s'))}{\|f_{\text{embd}}(l_{\text{desc}}(s))\| \|f_{\text{embd}}(l_{\text{desc}}(s'))\|}. \tag{2}$$

However, to be considered as a pseudometric, a function $d : \mathcal{S} \times \mathcal{S} \to \mathbb{R}$ must satisfy three conditions: $\forall x, y, z \in \mathcal{S}$, (i) $d(x, x) = 0$, (ii) $d(x, y) = d(y, x)$, and (iii) $d(x, z) \leq d(x, y) + d(y, z)$. In our case, $d_{\text{lang}}$ satisfies conditions (i) and (ii), but does not meet the condition (iii), the triangle inequality.

To overcome this problem, we propose to impose the constraint in Eq. 1, only for adjacent pairs of states $\forall (s, s') \in \mathcal{S}_{\text{adj}}$, where two states $s$ and $s'$ are considered adjacent if $s'$ can be reached from $s$ within a single state transition. Therefore, every encountered transition sample of $(s, s')$ during training is adjacent. We claim that this modification to Eq. 1 will implicitly induce a valid pseudometric $\tilde{d}$ for all state pairs $\forall (s, s') \in \mathcal{S}$.

**Claim 1.** *For any function $d_{lang} : \mathcal{S} \times \mathcal{S} \to \mathbb{R}_+$, there exists a valid pseudometric $\tilde{d} : \mathcal{S} \times \mathcal{S} \to \mathbb{R}_+$ that imposing the Eq. 1 on adjacent pairs of states lead to imposing the same equation on every possible state pairs under $\tilde{d}$, i.e.,*

$$\forall x, y \in S_{adj}, \ \|\phi(x) - \phi(y)\| \leq d_{\text{lang}}(x, y) \implies \forall x, y \in S, \ \|\phi(x) - \phi(y)\| \leq \tilde{d}(x, y).$$

We provide a proof of the claim in Appendix C. Applying this modification, we came to the final objective function as follows:

$$\mathcal{J}_{\text{LGSD}} = \mathbb{E}_{\pi, z \sim p(Z)} \left[ \sum_{t=0}^{T-1} (\phi(s_{t+1}) - \phi(s_t))^T z \right], \ s.t. \ \forall (s, s') \in \mathcal{S}_{\text{adj}}, \ \|\phi(s') - \phi(s)\| \leq d_{\text{lang}}(s, s').$$

Consequently, this objective encourages agents to explore diverse and distant states in latent space, resulting in a greater language-distance traveled along the trajectory within the state space.

## 4.4 UTILIZING LEARNED SKILLS USING NATURAL LANGUAGE

After training is complete, we need a way of selecting specific $z$ values to utilize the behaviors associated with each skill to solve concrete tasks. However, we lack knowledge about which $z$ leads to the desired behavior. We can sweep over the skill space at fixed intervals (Laskin et al.,

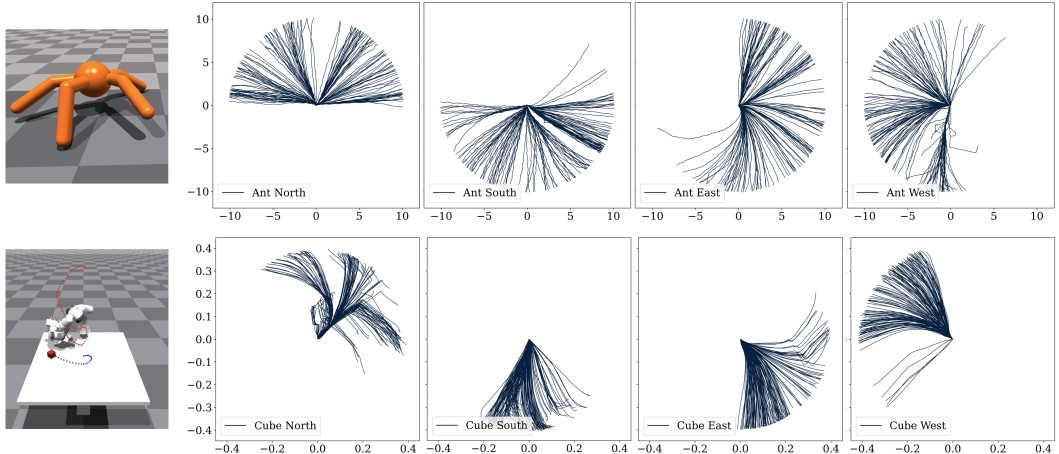

Figure 4: Trajectories of different skills trained with different prompts. For the Ant (top row), we recorded the base's $x, y$ coordinates. For the Franka robot-arm agent (bottom row), we recorded the $x, y$ coordinates of the object on the table.

2022), or sample a number of skills and evaluate each with hundreds of rollout trajectories (Park et al., 2023a). To overcome these challenges, Park et al. (2021; 2023b) have suggested using $z = (\phi(g) - \phi(s))/|\phi(g) - \phi(s)|_2$ for continuous skills, when we have Lipschitz continuous $\phi$. Yet, in many cases, we cannot fully specify the desired goal state. For instance, consider an agent using a robotic arm to move a cube. The state includes the end-effector's rotation and position, as well as the cube's coordinates. While you can specify the cube's position, the required rotation and position of the end-effector to achieve that position may not be known.

To address the challenge of goal state specification, we propose training a separate network $\psi : f_{\text{embd}}(l_{\text{desc}}(s)) \to Z$ to infer the skill. Intuitively, $\psi$ takes the embedding of the language description of the state $s$ as input and infers which $z$ was used to reach that specific state. Since we are gathering pairs of state $s$ and the corresponding skill $z$ during the training of $\pi$, we can train $\psi$ using the same data. After training, we can use $\psi$ and $s_{\text{goal}}$ to produce $z_{\text{goal}}$, which enables the agent to reach $s_{\text{goal}}$. We then feed $z_{\text{goal}}$ into $\pi$, allowing the agent to reach the goal state as described in natural language in a zero-shot manner. The full algorithm, including the training of $\psi$, is presented in Appendix E.

## 5 EXPERIMENTS

In this section, we evaluate our proposed LGSD by conducting a series of experiments on continuous control environments, encompassing both locomotion and manipulation setups. We aim to answer four questions: (1) Can prompting constrain the skill space into a desired semantic subspace? (2) Can language guidance lead to obtaining more diverse skills compared to unsupervised skill discovery baselines? (3) Can we utilize learned skills for solving downstream tasks? (4) Can we employ learned skills using natural language?

**Experimental setup** We trained our algorithm and baselines using Isaac Gym (Makoviychuk et al., 2021), a high-throughput GPU-based physics simulator. For the language model, we employed `gpt-4-turbo-2024-04-09`(Achiam et al., 2023). We set the temperature parameter of the language model to 0 to get a consistent, low-variance measure of $d_{\text{lang}}$. To reduce the number of unique queries, we discretized states and cached the input and output of these queries and reused them during training. We provide the exact prompts used for each experiments in Appendix G.

### 5.1 CONSTRAINING THE SKILL SPACE INTO A DESIRED SEMANTIC SUBSPACE

We first explored how LGSD constrains the skill search space into a desired semantic subspace using prompts. To visually present this concept, we designed experiments where agents visit mutually exclusive states depending on the prompts. We tested this idea in two environments, `Ant` and

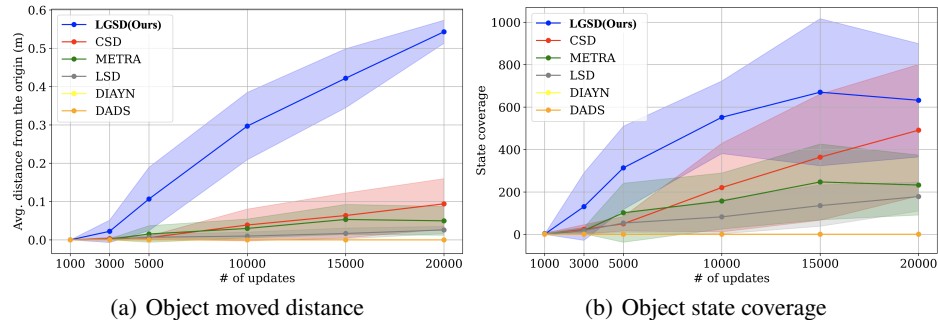

(a) Object moved distance          (b) Object state coverage

Figure 6: Comparison of the object's moved distance and state coverage by LGSD against baselines. LGSD moved the object (a) further and toward (b) more diverse directions. Additionally, it is more sample-efficient with the aid of language guidance.

`FrankaCube`. In the `Ant` environment, we required the agent to traverse only half of the plane. For instance, we allowed an agent to visit any location in the northern part of the plane but restricted it from entering the southern area. We repeated these experiments for each cardinal direction: North, South, East, and West (NSEW). Similarly, in the `FrankaCube` environment, we intended for the robot arm to move the object to only half of the plane for each of the NSEW directions starting from the origin. We trained four agents, each with its own unique prompt, encouraging the agent to explore only within the intended semantic subspace.

Fig. 4 (and Fig. 10 in Appendix) shows the discovered skills trained with each prompt. For each plot, we sampled $z \sim \mathcal{N}(0, I)$ 150 times independently and then generated trajectories using a skill-conditioned policy. As shown in Fig. 4, LGSD effectively constrained the skill space into the desired subspace, especially in the `Ant` environment. For the `Franka` agent, the robot arm was able to move the object into the intended region, but it could not fully cover all skills within the constrained subspace. We believe this limitation is due to the inherent nature of the exploration problem. In the `Ant` environment, the dynamics of the agent is invariant to horizontal translation, which allows effective control at any location. On the other hand, moving the cube along a long trajectory often requires changes in pushing strategies with different contacts and force profiles.

## 5.2 LANGUAGE GUIDANCE AIDS DISCOVERING DIVERSE SKILLS

To observe the effect of language guidance, we evaluate the diversity measure of the skills learned using LGSD against unsupervised skill discovery baselines. For the baselines, we take two classical mutual information-based skill discovery algorithms: DIAYN (Eysenbach et al., 2018) and DADS (Sharma et al., 2019), as well as three state-of-the-art distance maximization based approaches: LSD (Park et al., 2021), CSD (Park et al., 2023a), and METRA (Park et al., 2023b).

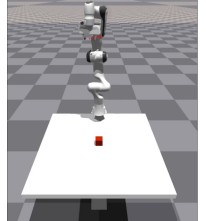

Although METRA was originally designed for pixel-based input tasks, it can also handle vector-based inputs. Therefore, we used vector-based state inputs for all methods to facilitate direct comparison. For all methods, we utilized no human prior knowledge in selecting parts of the state dimension to induce desired behaviors; instead, we passed full observations directly to the skill discovery algorithms. Detailed information about the observations is provided in Appendix F.2.1.

Figure 5: Initial state of robot arm.

We set up a challenging manipulation task as illustrated in Fig. 5. Here, we expect to discover skills that can manipulate the object toward diverse directions. The challenge arises from the initial distance between the robot arm and the object, which is set to the maximum possible value. We believe this task can confirm the validity of the two-stage prompt suggested in section 4.2. After training, we randomly sampled 200 skills and measured both the average moved distance of the object and the state coverage that counts the total number of visited horizontal grids, each sized 1cm x 1cm.

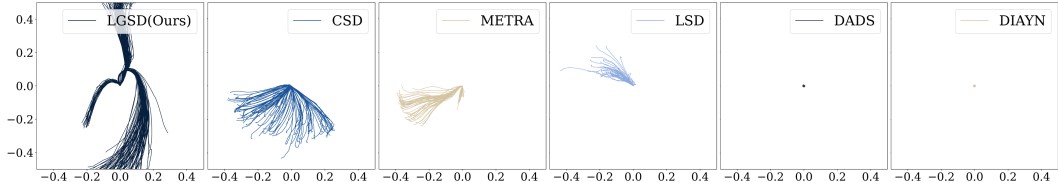

Figure 7: Paths of objects moved on a horizontal table by LGSD and five baseline algorithms, illustrated using the best-performing checkpoint from four runs for each algorithm.

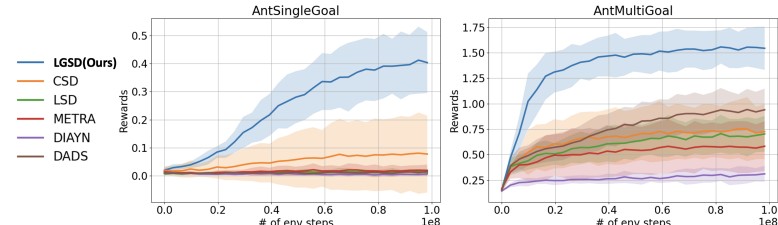

Figure 9: The training curve of the high-level controller in locomotion tasks, using the low-level controller trained with LGSD and five state-of-the-art unsupervised skill discovery methods. LGSD is able to achieve better returns and exhibits better sample efficiency.

We present the results in Fig. 6. As shown in Fig. 6, LGSD discovers more diverse states compared to all baseline algorithms. Skills trained using LGSD moved the object an average of 0.5m, which is five times greater than the value achieved by the second-best algorithm. This result confirms that language guidance from an LLM aids in discovering both diverse and meaningful skills. Notably, all distance-maximization methods (CSD, METRA, LSD) could manipulate the object, whereas MI-based methods could not. Fig. 7 shows qualitative results of the learned behaviors by each method, where LGSD demonstrated the ability to manipulate objects to cover more grid areas. Additionally, visualizations and an analysis of the agent's movements in latent space are provided in Appendix D.

## 5.3 HARNESSING LEARNED SKILLS

**Zero-shot task solving via natural language** We first demonstrate how our agent can solve a downstream task in a zero-shot manner via natural language instruction. As discussed in Section 4.4, we utilize $\psi(z|f_{\text{embed}}(l_{\text{desc}}))$ to infer a skill to reach the goal state. For instance, our instruction can be a description of the desired state, `"The object is located at [0.3, 0.2]"`, which does not contain information regarding the robot-arm's position or orientation. The learned $\psi$ allows an agent to translate the given language instruction into an effective skill for reaching the goal state.

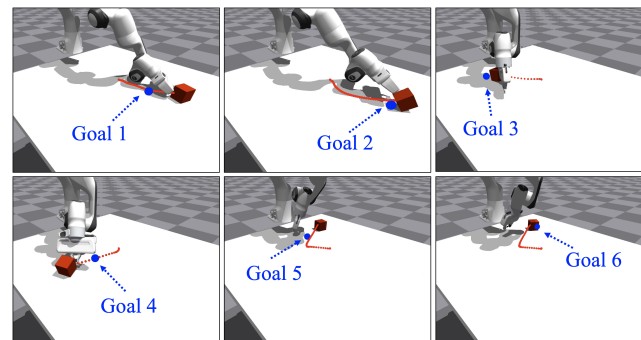

Figure 8: Users can specify goals in natural language. Blue dots represent the desired position of the object, while red dots indicate the object's trajectory.

Fig. 8 illustrates scenarios that a robot-arm moves the object to the six different goal positions described in natural language. This experiment showcased that the agent can understand the desired goal state with the help of $\psi$, execute the skill accurately, and successfully move the cube to the target positions in a zero-shot manner. We also provide the results of the zero-shot goal-following agent in the locomotion domain in Appendix, Fig. 11.

**Training high-level controller with learned skills**   Although our method enables training a desired set of skills to some extent using tailored prompts, it can be challenging for users to specify exact prompts that produce behaviors for more complex tasks. In such scenarios, our method can act as a pre-training module, which trains a low-level controller $\pi_{low}(a|s,z)$, which can then be leveraged by a high-level controller $\pi_{high}(z|s,g)$ to select the appropriate skills for new downstream task, represented by $g$.

We evaluate the performance of high-level controllers that leverage a pre-trained LGSD low-level controller on two downstream tasks: `AntSingleGoal` and `AntMultiGoal`. Details of the downstream tasks are available in Appendix F.1. Fig. 9 shows that by using the set of skills trained with our method, the high-level controller can achieve higher returns and better sample complexity compared to prior skill discovery methods. We hypothesize that these improvements comes from the fact that the skills discovered through LGSD can better focus on more semantically meaningful changes in the $x, y$ positions by leveraging guidance from the LLM, while other methods simply seek changes in all 39 dimensions of the states equally.

## 6   CONCLUSION

Our work introduced Language Guided Skill Discovery (LGSD), a novel framework for skill discovery that leverages the semantic understanding capabilities of large language models (LLMs) to guide the learning of semantically diverse skills. By incorporating language as a tool for both constraining the skill space and measuring semantic diversity, LGSD offers a novel approach to learning diverse skills. We have demonstrated through various experiments that LGSD not only constrains skills within semantically meaningful subspaces but also enhances the diversity and applicability of the skills learned.

Despite these advancements, there are areas for further exploration and improvement. The approach could benefit from a more nuanced understanding of the scene, potentially by incorporating vision-language models (VLMs). We conjecture that some semantics can be more easily captured through vision. VLMs would effectively help to utilize these dimensions. Furthermore, we suggest that incorporating trajectory-level semantic differences, instead of state-level, could be an interesting future direction. We could provide entire trajectories to LLMs/VLMs and query them to associate them with semantic meanings.

Overall, we believe our work marks the beginning of a series of efforts that capture semantic diversity between skills. We hope this work facilitates further research endeavors in discovering semantically meaningful skills with the aid of external knowledge sources, including LLMs.

**Reproducibility Statement**   We have made significant efforts to ensure the reproducibility of our work across various aspects.

- Detailed information about the observations used in our experiments is provided in Appendix F.2.1.

- A comprehensive pseudo-code of our algorithm is available in Appendix E.

- A rigorous proof of Claim 4.3 is included in Appendix C.

- Visualizations of the learned representation space can be found in Appendix D.

- Details of the high-level tasks training are provided in Appendix F.1.

- The exact version of the LLM used and all prompts are documented in Appendix G.

**Acknowledgements**   This research has been funded by the Industrial Technology Innovation Program (P0028404, development of a product-level humanoid mobile robot for medical assistance equipped with bidirectional customizable human-robot interaction, autonomous semantic navigation, and dual-arm complex manipulation capabilities using large-scale artificial intelligence models) of the Ministry of Industry, Trade and Energy of Korea.

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

APPENDIX

# A ADDITIONAL RESULT FIGURES

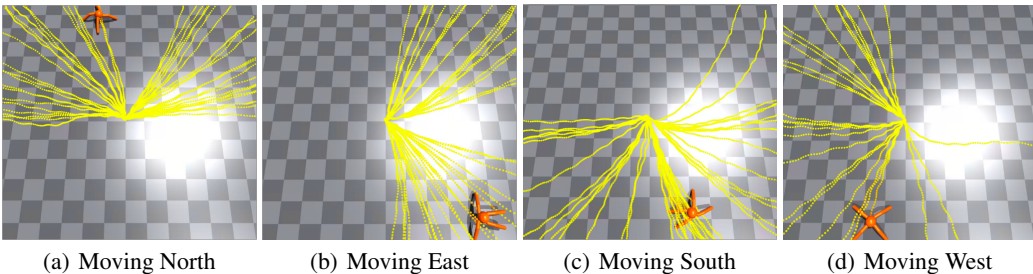

(a) Moving North     (b) Moving East     (c) Moving South     (d) Moving West

Figure 10: LGSD constrains different prompts into different semantic subspace of skills. Each yellow line indicates a trajectory generated using a randomly sampled skill learned with each prompt. Using different prompts, each set of skills is constrained to visit the (a)Northern, (b)Eastern, (c)Southern, and (d)Western areas of the plane.

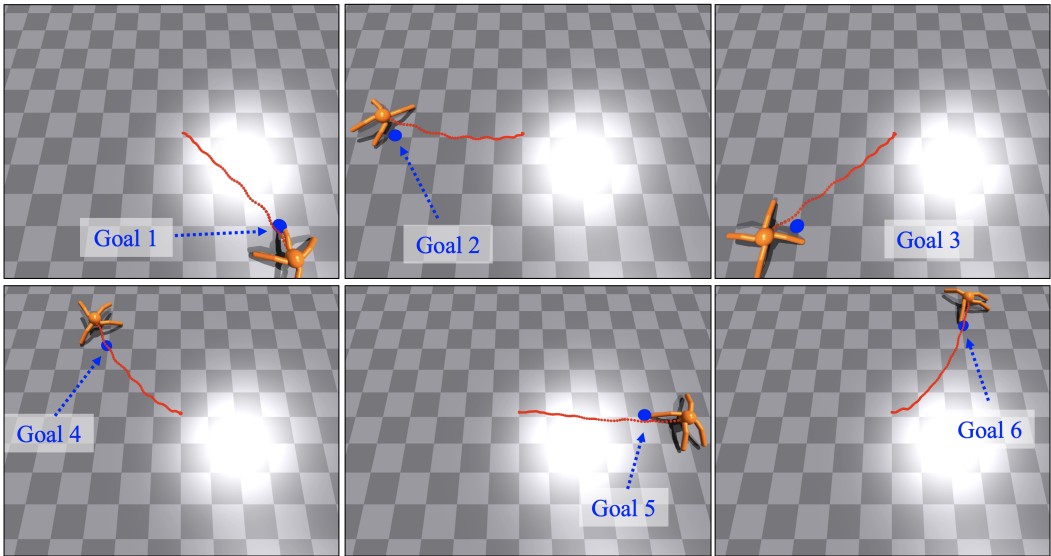

Figure 11: Ant agents move toward the goal. Red dots indicate the actual path of the agent's base, while the blue dot represents ths goal. Each goal is specified using natural language.

# B PROOF OF MAXIMIZING WASSERSTEIN DEPENDENCY MEASURE

We begin with the alternative form of Wasserstein distance using Kantorovich-Rubenstein duality following Villani et al. (2009), to make $\mathcal{I}_{\mathcal{W}}$ tractable.

$$\mathcal{I}_{\mathcal{W}}(S;Z) = \sup_{f \in \mathcal{L}_{S \times Z}} \mathbb{E}_{p(S,Z)}[f(S,Z)] - \mathbb{E}_{p(S)p(Z)}[f(S,Z)]$$

where $\mathcal{L}_{S \times Z}$ is a set of all 1-Lipschitz function defined in $S \times Z \to \mathbb{R}$ under the distance metric $d$. For the remainder, we follow the proof outlined in Park et al. (2023b), but we provide a simplified version.

We first parameterize $f(S, Z) = \phi(S)^T \psi(Z)$ using $\phi : \mathcal{S} \to \mathbb{R}^D$ and $\psi : \mathcal{Z} \to \mathbb{R}^D$, which are 1-Lipschitz constrained functions. This is safe decomposition because the expressive power of $\phi(S)^T \psi(Z)$ is universal. Please refer to Park et al. (2023b) for more detail.

$$\mathcal{I}_{\mathcal{W}}(S; Z) \approx \sup_{\|\phi\|_L \leq 1, \|\psi\|_L \leq 1} \mathbb{E}_{p(S,Z)}[\phi(S)^T \psi(Z)] - \mathbb{E}_{p(S)}[\phi(S)]\mathbb{E}_{p(Z)}[\psi(Z)]$$

Then we use $Z$ instead of $\psi(Z)$ to earn simplicity at the cost of expressiveness. Also we choose the prior distribution $p(Z)$ to have zero-mean. This makes the second term $\mathbb{E}_{p(S)}[\phi(S)]\mathbb{E}_{p(Z)}[Z]$ to 0. Finally, similar to VIC, we measure WDM between the final state $S_T$ and the skill, insteaf of all states.

$$\mathcal{I}_{\mathcal{W}}(S_T; Z) \approx \sup_{\|\phi\|_L \leq 1} \mathbb{E}_{p(S,Z)}[\phi(S_T)^T Z]$$

Next, we utilize $\mathbb{E}_{p(S,Z)}[\phi(S_0)^T Z] = \mathbb{E}_{p(S)}[\phi(S_0)]\mathbb{E}_{p(Z)}[Z] = 0$ because the distribution of the initial state $S_0$ and $p(Z)$ are mutually independent, and $p(Z)$ has a zero-mean. So subtracting it from the right-hand side doesn't change the equation. Now we have the final objective:

$$\mathcal{I}_{\mathcal{W}}(S_T; Z) \approx \sup_{\|\phi\|_L \leq 1} \mathbb{E}_{p(S,Z)}[\phi(S_T)^T Z] - \mathbb{E}_{p(S,Z)}[\phi(S_0)^T Z]$$

$$\approx \sup_{\|\phi\|_L \leq 1} \mathbb{E}_{p(\tau,z)} \left[ \sum_{t=0}^{T-1} (\phi(S_{t+1}) - \phi(S_t))^T Z \right].$$

Therefore, we can use RL with the reward function $r(s, z, s') = (\phi(s') - \phi(s))^T z$ to maximize $\mathcal{I}_{\mathcal{W}}(S_T; Z)$.

## C  PROOF OF CLAIM 1

Our goal is to prove the following two claims.

(i)  $\forall(x, y) \in S_{adj}, \|\phi(x) - \phi(y)\| \leq d_{\text{lang}}(x, y) \implies \forall x, y \in S, \|\phi(x) - \phi(y)\| \leq \tilde{d}(x, y)$

(ii)  $\tilde{d}(x, y)$ is a valid pseudometric.

We follow the proof of CSD (Park et al., 2023a) similarly, but it has significant differences which will be covered in later of this section.

**Proof of (i)**  We begin with introducing $P(x, y)$ as the set of all finite path from state $x$ to state $y$, for $\forall x, y \in \mathcal{S}$. Here, we only consider *existing* path $p = (s_0 = x, s_1, ..., s_{T-1}, s_T = y) \in P(x, y)$. The term *existing* indicates that for every consecutive state pairs $(s_i, s_{i+1})$ in $p$, $s_{i+1}$ should be reachable from $s_i$. More concretely,

$$\forall(s_i, s_{i+1}) \in p, \ \exists a \in \mathcal{A}, s.t., \mathcal{P}(s_i, a, s_{i+1}) > 0$$

where, $\mathcal{P}$ is a transition probability. Then, for a path $p$, we denote the sum of the language distance along the path $p$ as $D_{\text{lang}}(p)$, i.e., $D_{\text{lang}}(p) \stackrel{\text{def}}{=} \sum_{t=0}^{T-1} d_{\text{lang}}(s_t, s_{t+1})$. Now we can define $\tilde{d}(x, y)$ as follows:

$$\tilde{d}(x, y) \stackrel{\text{def}}{=} \begin{cases} \inf_{p \in P(x,y)} D_{\text{lang}}(p) & \text{if } x \neq y \\ 0 & \text{if } x = y. \end{cases}$$

Now we are ready to move on to the main proof. We assume that $\forall(x, y) \in S_{adj}, \|\phi(x) - \phi(y)\| \leq d_{\text{lang}}(x, y)$. Also, note that $\forall x, y \in \mathcal{S}$, both $d_{\text{lang}}(x, x) = 0$ and $d_{\text{lang}}(x, y) = d_{\text{lang}}(y, x)$ holds

trivially. We denote the optimal path as $p^* = (x, s_1^*, s_2^*, ..., s_{T-1}^*, y) \in P(x, y)$ which satisfy $\tilde{d}(x, y) = D_{\text{lang}}(p^*)$. Then,

$$
\begin{aligned}
\forall x, y \in \mathcal{S}, \ \|\phi(x) - \phi(y)\| &= \|\phi(x) - \phi(s_1^*) + \phi(s_1^*) - \phi(s_2^*) + ... + \phi(s_{T-1}^*) - \phi(y)\| \\
&\leq \|\phi(x) - \phi(s_1^*)\| + \|\phi(s_1^*) - \phi(s_2^*)\| + ... + \|\phi(s_{T-1}^*) - \phi(y)\| \\
&\leq d_{\text{lang}}(x, s_1^*) + d_{\text{lang}}(s_1^*, s_2^*) + ... + d_{\text{lang}}(s_{T-1}^*, y) = \tilde{d}(x, y).
\end{aligned}
$$

This concludes the proof of (i).

**Proof of (ii)** To show that $\tilde{d}$ is a valid pseudometric, we need to prove following three conditions: $\forall x, y, z \in \mathcal{S}, \tilde{d}(x, x) = 0, \tilde{d}(x, y) = \tilde{d}(y, x)$, and $\tilde{d}(x, y) \leq \tilde{d}(x, z) + \tilde{d}(z, y)$.

At first, given that $\forall x \in \mathcal{S}, d_{\text{lang}}(x, x) = 0$ , $\tilde{d}(x, x) = 0$ holds trivially.

Secondly, given that $\forall x, y \in \mathcal{S}, d_{\text{lang}}(x, y) = d_{\text{lang}}(y, x)$ , $\tilde{d}(x, y) = \tilde{d}(y, x)$ also holds trivially.

For the triangle inequality of $\tilde{d}$,

$$
\begin{aligned}
\forall x, y, z \in \mathcal{S}, \ \tilde{d}(x, y) &= \inf_{p \in P(x,y)} D_s(p) \\
&\leq \inf_{p_1 \in P(x,z), p_2 \in P(z,y)} D_s(p_1) + D_s(p_2) \\
&= \inf_{p_1 \in P(x,z)} D_s(p_1) + \inf_{p_2 \in P(z,y)} D_s(p_2) \\
&= \tilde{d}(x, z) + \tilde{d}(z, y).
\end{aligned}
$$

This completes the proof of (ii).

**Difference from CSD** The primary distinction is that we focus exclusively on the *existing* path $p \in P(x, y)$ only. Here, we refer a path "exists" between state $x$ and $y$ only if state $y$ is reachable from $x$ within a finite number of transitions. During the actual optimization process, our agents encounter existing path only, so we believe this constraints align our theory more closely with the actual training scenario. Due to this specific focus, unlike from CSD, $\tilde{d}$ can no longer serves as a lower bound for $d_{\text{lang}}$.

Another notable difference is that we update only with the adjacent pairs of states. Thanks to our claim 1., we still can induce a valid pseudo-metric across all pairs of states within the state space. This approach aims to bridge the gap between our theoretical model and the training data used for model updates.

# D LATENT SPACE VISUALIZATION

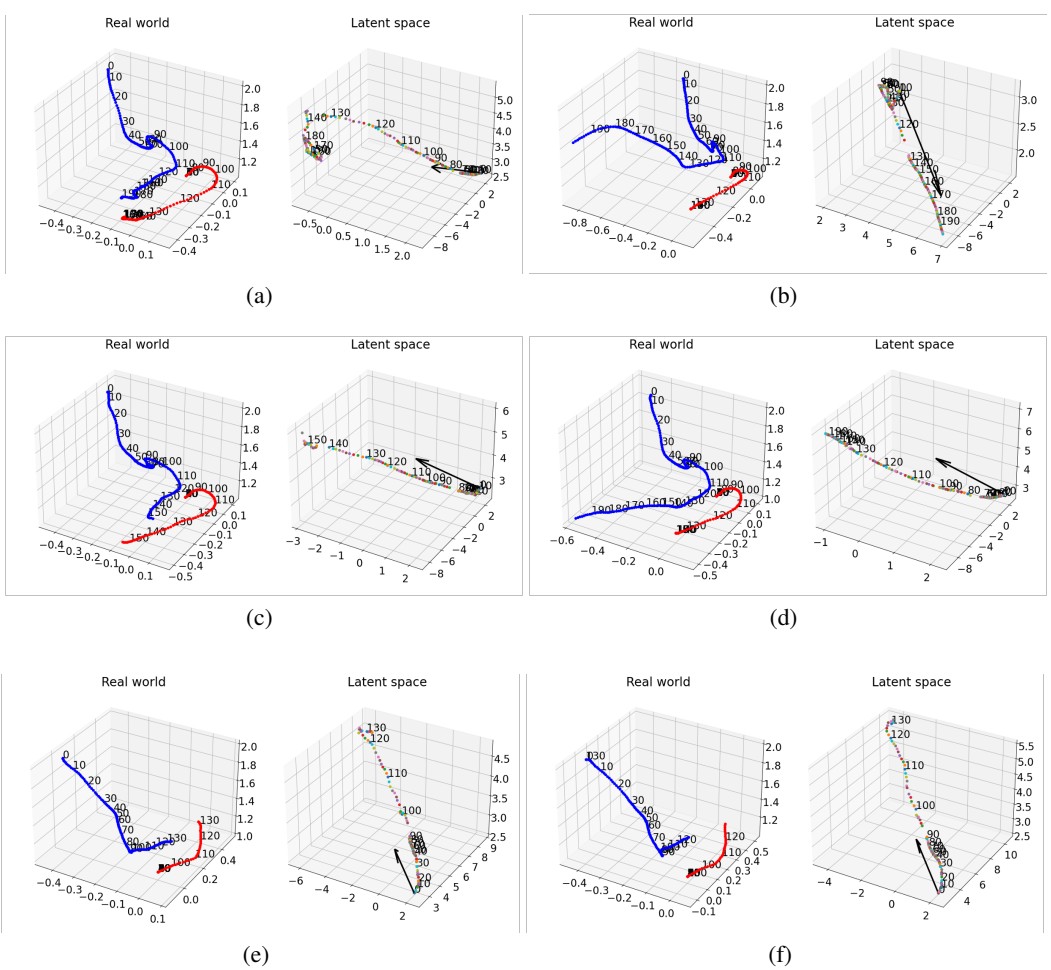

Figure 12: Each sub-figure corresponds to a trajectory drawn from a policy conditioned on six different skills sampled randomly. In each subfigure, the **left** figure displays the actual paths of both the end-effector (blue) and the object (red) in the real world while the **right** figure illustrates the corresponding trajectory in latent space, generated using the learned $\phi$ function. A black arrow indicates the skill vector used to produce the trajectory. The numbers indicate transition step in each of the episode.

As discussed in Section 4.1, LGSD encourages the agent to spread out evenly in the latent space. The distance between two consecutive points in the latent space is regulated by the language distance, $d_{\text{lang}}$. Fig. 12 demonstrates that the orientation of the agent's trajectory in latent space aligns well with the sampled skill vector. Since the skill vector is sampled isotropically, agent effectively spread out in the latent space. Additionally, it is noteworthy that the distance between two consecutive points in the latent space increases especially when the cube is moved. On the other hand, when the cube is stationary, there is no further extension in the latent space as can be seen in the Fig. 12 (a). This indicates that our language distance $d_{\text{lang}}$ serves as a meaningful proxy for semantic distance.

# E FULL ALGORITHM OF LGSD

---

**Algorithm 1** Language Guided Skill Discovery

---

1: Initialize skill-conditioned policy $\pi(a|s, z)$, representation function $\phi(s)$, prompt $l_{\text{prompt}}$, LLM function $LLM$, language embedding model $f_{\text{embed}}$, skill inference network $\psi$, Lagrange multiplier $\lambda$, and data buffer $\mathcal{D}$
2: **for** $i \leftarrow 1$ to # of epochs **do**
3:     **for** $j \leftarrow 1$ to # of episodes per epoch **do**
4:         Sample skill $z \sim \mathcal{N}(0, I)$
5:         **while** episode not terminates **do**
6:             Sample action $a \sim \pi(a|s, z)$
7:             Execute $a$ and receive $s'$
8:             Query $LLM(\cdot|s, l_{\text{prompt}})$ to produce $l_{\text{desc}}(s)$ and $l_{\text{desc}}(s')$
9:             Compute reward $r = (\phi(s') - \phi(s))^T z$
10:           Compute $d_{\text{lang}}(s, s')$ using eq. (2)
11:           Compute embedding vector $e_s = f_{\text{embed}}(l_{\text{desc}}(s))$
12:           Add $\{s, a, r, s', d_{lang}(s, s'), e_s, z\}$ to buffer $\mathcal{D}$
13:         **end while**
14:     **end for**
15:     **for** $\{s, a, r, s', d_{lang}(s, s'), e_s, z\}$ in $\mathcal{D}$ **do**
16:         Update $\phi$ to maximize $\mathbb{E}_{(s,z,s') \sim \mathcal{D}} \left[ (\phi(s') - \phi(s))^T z + \lambda \cdot \min(\epsilon, d_{\text{lang}}(s, s') - \|\phi(s) - \phi(s')\|_2^2) \right]$

17:         Update $\lambda$ to minimize $\mathbb{E}_{(s,z,s') \sim \mathcal{D}} \left[ \lambda \cdot \min(\epsilon, d_{\text{lang}}(s, s') - \|\phi(s) - \phi(s')\|_2^2) \right]$
18:         Update $\pi$ using PPO with reward $r$
19:         Update $\psi$ to minimize Mean_Squared_Error between $\psi(e_s)$ and $z$
20:     **end for**
21: **end for**

---

# F IMPLEMENTATION DETAILS

## F.1 DETAILS OF DOWNSTREAM TASKS

We used similar settings for the downstream tasks in the locomotion domain as those used in Metra (Park et al., 2023b). For `AntSingleGoal`, a single goal is sampled for every episode within a range of $[-50, 50]$, centered at the origin. The task is considered complete when the agent reaches within a radius of 3 units around the goal, at which point the agent receives a reward of 1 and the episode terminates. For `AntMultiGoal`, there are four sequential goals in total. When the first goal is reached, the next goal is sampled randomly within a range of $[-7.5, 7.5]$, centered at the agent's current location. Reaching each goal gives a reward of 1.

On the other hand, we have two differences from the task used in Metra. The first difference is in the termination condition. When the agent's torso height drops below the threshold height of 0.31 m, the episode is instantly terminated. This threshold is the default value used in the Ant environment from the official IsaacGymEnvs library. We believe this condition encourages the agent to walk rather than "roll", which often occurs when attempting to maximize all possible state dimensions without constraints. Another difference is that we use a higher-dimensional observation space, as detailed in F.2.1. Our observation space has 39 dimensions, whereas the original Metra work has 26 dimensions, as it is based on the MuJoCo Ant environment.

We trained the low-level controller five times with different seeds for reporting final performance. We used 20,000 updated checkpoints from each training runs of low-level controller to train the high-level controller.

## F.2 OBSERVATIONS AND HYPERPARAMETERS

### F.2.1 OBSERVATIONS

Table 1: Ant Environment Observations

| Name | Description | Dimension |
|---|---|---|
| Base position | x,y,z position of the robot's base | 3 |
| Base velocity | velocity of robot's base in x,y,z direction | 3 |
| Base angvel | angular velocity of robot's base | 3 |
| Base rotation | Yaw, Pitch, Roll of robot's base | 3 |
| Gravity projection | Vector indicates direction of the gravity | 3 |
| DOF position | Current angle of each DOF | 8 |
| DOF velocity | Angular velocity of each DOF | 8 |
| Previous action | Action executed in previous step | 8 |
| Sum | | 39 |

Table 2: Franka Manipulator Environment Observations

| Name | Description | Dimension |
|---|---|---|
| Cube pos | x,y,z coordinate of the target cube | 3 |
| Cube quat | Rotation of the target cube | 4 |
| EEF pos | x,y,z coordinate of robot arm's end-effector | 3 |
| EEF quat | Rotation of the Robot arm's end-effector | 4 |
| Gripper q | Gripper width | 2 |
| Sum | | 16 |

### F.2.2 HYPERPARAMETERS

Table 3: Hyperparameters of LGSD

| Name | Value |
|------|-------|
| Learning rate | 0.0001 |
| Optimizer | Adam(Kingma & Ba, 2014) |
| Minibatch size | 32768(Ant) , 16384(Franka) |
| Horizon length | 32 |
| PPO clip threshold | 0.2 |
| PPO number of epochs | 5 |
| GAE $\lambda$ (Schulman et al., 2015) | 0.95 |
| Discount factor $\gamma$ | 0.99 |
| Entropy coefficient | 0.0001 |
| Initial Lagrange coefficient $\lambda$ | 300 |
| Dim. of skill $z$ | 2(Ant), 3(Franka) |
| Policy network $\pi$ | MLP with [256, 256, 128], |
| Activaion of $\pi$ | ELU(Clevert et al., 2015) |
| Representation function $\phi$ | MLP with [256, 256, 128] |
| Activaion of $\phi$ | ReLU |
| Skill inference network $\psi$ | MLP with [256, 256, 128] |
| Activaion of $\psi$ | ReLU |

## G  PROMPTS USED FOR EACH OF THE EXPERIMENTS

In this section, we share the prompts used for each of the experiments, and the examplary outputs from the LLM. One key detail is that we prompted LLM to finalize its sentence with a keyword of `Description:`, so that we can parse the final description out of whole generated sentences.

### G.1  PROMPTS FOR ENFORCING ANT AGENT TO MOVE NORTH

**Input**  `You will describe a given robot's state where the robot is moving on a plane.  The state of the robot is as follows:`

`Robot's x, y, z coordinates:  [6.5, -0.5, -0.3] Robot's rotation: [0.9770, 0.1942, 0.0884, 0.0012]`

`If the y-coordinate is less than 0, we consider the area as southern.  Conversely, if the y-coordinate of the robot is greater than or equal to 0, we consider the area as northern.`

`If the robot is in the northern area relative to the origin, provide the robot's [x, y] position.  Please conclude your answer with the final description in the format:  'Description:  The robot's x, y position is ___'.`

`If the robot is in the southern area relative to the origin, just state that 'Description:The robot is located in the southern area.'`

**Output**  `Since the y-coordinate of the robot is -0.5, which is less than 0, the robot is located in the southern area.`

`Description:  The robot is located in the southern area.`

### G.2  PROMPTS FOR ENFORCING FRANKA AGENT TO PUSH OBJECTS TOWARD NORTH

**Input**  `You will describe a scene where a robot arm is approaching an object.  The state of the robot arm is as follows:`

```
Robot's end effector's x, y, z coordinates: [6.5, -0.5, -0.3],
Object's x, y, z coordinates: [0, 1.0, 0.2], Origin's x, y, z
coordinates: [0, 0, 0]
```

If the object's y coordinate is less than 0, tell me 'Description: The object is located in southern area.'

If the object's y coordinate greater than or equal to 0, tell me the object's x, y coordinates. You can use any introductory words, but please conclude your answer with the final description in the format: 'Description: The object's x, y position is ___.'

If the object's coordinates match the origin's, tell me the distance between the robot's end effector and the object. Again, conclude with the final description in the format: 'Description: the distance is ___ units.'

**Output1 - when object is moved** In this scene, the object's y coordinate is 1.0, which is greater than 0. Thus, the x and y coordinates of the object are 0 and 1.0 respectively.

Description: The object's x, y position is 0, 1.0.

**Output2 - When object is not moved** In this scene, we observe a robot's end effector positioned at coordinates [6.5, -0.5, -0.3] aiming to approach an object. The object itself is found at the coordinates [0, 0, 0], which are identical to the origin's coordinates.

To analyze this setup, we calculate the distance between the robot's end effector and the object. Using the Euclidean distance formula:

$$\text{Distance} = \sqrt{(x_2 - x_1)^2 + (y_2 - y_1)^2 + (z_2 - z_1)^2}$$
$$= \sqrt{(0 - 6.5)^2 + (0 - (-0.5))^2 + (0 - (-0.3))^2}$$
$$= \sqrt{6.5^2 + 0.5^2 + 0.3^2}$$
$$= \sqrt{42.25 + 0.25 + 0.09}$$
$$= \sqrt{42.59}$$
$$\approx 6.53 \text{ units}$$

Description: The distance is 6.53 units.

### G.3 PROMPTS FOR FRANKA AGENT MANIPULATES OBJECT IN GENERAL

**Input** You will describe a scene where a robot arm is approaching an object. The state of the robot arm is as follows:

```
Robot's end effector's x, y, z coordinates: [6.5, -0.5, -0.3],
Object's x, y, z coordinates: [0, 1.0, 0.2], Origin's x, y, z
coordinates: [0, 0, 0]
```

If the object's coordinates differ from the origin's, tell me the object's x, y coordinates. You can use any introductory words, but please conclude your answer with the final description in the format: 'Description: The cube's x, y position is ___.'

If the object's coordinates match the origin's, tell me the distance between the robot's end effector and the object. Again, conclude with the final description in the format: 'Description: the distance is ___ units.'

**Output** In the scene, a robot arm's end effector is positioned at coordinates [6.5, −0.5, −0.3], while an object is located at [0, 1.0, 0.2]. The origin of the coordinate system is at [0, 0, 0]. Since the object's coordinates do not match the origin's coordinates, we focus on the object's position relative to the origin.

Description: The object's x, y position is [0, 1.0].

