# OpenReview forum: "Language Guided Skill Discovery"
_ICLR.cc/2025/Conference — ICLR 2025 Poster_

### Official Review · Reviewer_BQeq · 2024-11-03

**Soundness:** 3
**Presentation:** 3
**Contribution:** 3
**Rating:** 8
**Confidence:** 4

**Summary:**

This paper presents a novel skill discovery framework called Language Guided SKill Discovery (LGSD). LGSD aims to maximize the semantic diversity of discovered skills by utilizing language guidance from LLMs. In LGSD, the interactions between human experts and LLMs are utilized for 1) constraining the skill search space into a semantically desired subspace, and 2) encouraging the agent to visit semantically diverse states within the subspace. Various experiments are conducted to demonstrate the capability of the LGSD framework to find more diverse and distinctive skills compared to previous methods.

**Strengths:**

- This paper is pretty clear wrt the high-level topic and what the high-level intuitions are. The idea of incorporating LLMs to guide the skill discovery is novel and intuitive. Figure 2 is a very clear figure except one minor issue (see weakness).
- By introducing the representation function \phi which transform the state space s to latent space z, the skill discovery problem in the Euclidean space is transformed to an equivalent problem in the latent space. Figures in Appendix D show that identifying skills in the latent space is much easier than in the original space. Furthermore, by formulating the reward as $r(s, z, s^\prime) = ( \phi(s) - \phi(s^\prime))^Tz$, LGSD successfully transform the problem of finding diverse skills to a problem of maximizing the culmulative designed reward.
- The section 2.1 is a great section that summarizes works related to unsupervised skill discovery.  Researchers might already read some related articles, but such a classification (definition)  as "mutual information based " and "distance maximization" approaches is still helpful to the whole community yet very accurate.
- Experiments results show LGSD is effective and competitive in discovering semantically distict skills compared to other skill discovery baselines.

**Weaknesses:**

- Figure 1 shows a set of discovered semantically distinctive skills but the caption is not very informative. It could be better if description of each skill can be presented.
- Figure 2 is very good. However, it is unlikely for reader to understand $d_{lang}(s, s^\prime)$ without any additionally context. People may mistakenly treat this as some Euclidean distance between s and s^prime in the original space until they encounter equation (2). I would like to see some quick explanation that the $d_{lang}$ here happens in latent space, and this procedure requires LLMs.
- Prompts are required in this work to interact with LLMs to acquire the semantic knowledge. The design of the prompt is critical to the direction or subspace of skill discovery. However, what if we only want to leverage the semantic knowledge from LLMs without setting up to much language prior? From my understanding, too much control from human means less diverse and improvising skills.
- This is minor. In Line 319, it is unclear what is $s$ and $g$ without defining them.

**Questions:**

- Why enforcing $\phi$ to be 1-Lipschitz continuous under $d_{lang}$? Is there any other way to constrain the difference in the latent space?
- In Section 5.1. In FrankCube environment, the intended half plane is only for NSEW directions. Can we see arbitrary directions, e.g., north-east or south-west?

---

> ### Author Response · Authors · 2024-11-21
>
> Thank you for your thoughtful review. Below, we provide detailed answers to your comments.
>
> **(W3) Less prior knowledge**
>
> Yes, we agree with your perspective. When the scope of potential downstream tasks is known, human-specified prompts can serve as a valuable prior, constraining the search space to skills relevant to those tasks. However, as you noted, true diversity in skills emerges when working with zero-prior settings. The ultimate goal of skill discovery research is to obtain general skills that can be applied to any type of downstream task. In such cases, the prompt could be as simple as just *“describe the scene,”* which minimizes reliance on priors. We see reducing priors to achieve more diverse behaviors as a promising direction for future work.
> \
> \
> **(Q1) Why enforcing 1-Lipschitz continuity for $\phi(s)$**
>
> The language distance $d_{lang}(s,s')$ is defined within the state space. We want a function $\phi:s \rightarrow z$ that maps state $s$ into latent space $z$, but want the function $\phi(s)$ to be regulated by the language distance $d_{lang}(s,s')$. Enforcing Lipschitz continuity is a common technique to achieve this property. For more details, please refer to:
>
> [1] Seohong Park, Oleh Rybkin, and Sergey Levine. Metra: Scalable unsupervised rl with metric-aware abstraction. In The Twelfth International Conference on Learning Representations, 2023.
>
> [2] Seohong Park, Kimin Lee, Youngwoon Lee, and Pieter Abbeel. Controllability-aware unsupervised skill discovery. In International Conference on Machine Learning, pp. 27225–27245. PMLR, 2023.
> \
> \
> **(Q2) Cube experiments with NorthEast, SouthWest**
>
> We conducted experiments on this aspect, and the results were successful. Please refer to the attached video in the supplementary material for more details.
> \
> \
> **(W1,2,4) Minor Corrections**
> - We will replace Fig. 1 with the "edible objects manipulation" tasks.
> - We will incorporate your feedback to improve Fig. 2.
> - In Line 319, we will provide additional elaboration on $s$ and $g$.

---

### Official Review · Reviewer_Dsd7 · 2024-11-03

**Soundness:** 3
**Presentation:** 4
**Contribution:** 2
**Rating:** 8
**Confidence:** 4

**Summary:**

“Language Guided Skill Discovery” proposes a method for learning diverse skills with unsupervised reinforcement learning by aligning skills with latent space transitions under a distance constraint. The novelty in the proposed method stems from the use of language distance as the constraint. This is done by a Large Language Model (LLM) which describes the robot state, and a language embedding model which encodes this description in a fixed real-valued vector where the distance between descriptions can be measured. The results show successful skill discovery on locomotion and robot manipulation environments.

**Strengths:**

- Interesting and novel contribution - while using LLMs is not rare, this is an interesting way of using them for encouraging skill discovery.

- The proposed method is well-supported mathematically.

- Good zero-shot language-specified goal tracking with the learned skill predictor network.

**Weaknesses:**

- Not fully convinced that LLMs are necessary for the proposed scenarios. Showing the performance on more complex situations where the semantic understanding of LLMs is necessary would make the case much stronger.

- More qualitative evaluation (i.e. videos) would be good for evaluating the learned skills.

**Questions:**

## Questions:
1. What is the dimensionality of the skill-space here? Is there any benefit to having higher dimensions / are more skills learned?

2. The performance of other skill discovery methods seems quite poor on downstream Ant tasks, which is surprising especially for LSD/CSD/METRA. Are the tasks or rewards defined here differently from their original papers?

3. Is the LLM necessary given those prompts? Could these examples not be done just as well with simple logic gates? For example, prompt G.3 could be defined as “if object not at origin, return position of object, else return distance between end effector and object” without any LLM.
I understand that this is more or less a proof of concept, but I think it would go a long way if you can show some examples where the “common knowledge” of LLMs can be properly utilized.


4. I am curious if specifying just the object’s Cartesian position as the sole encoder observation under a Euclidean distance constraint (LSD) would yield similar results. I guess it might struggle more with exploration (no signal before the end-effector has collided with the object), but it would be interesting to see.


5. I like the visualizations (and analysis) in Appendix D. As I understand, the language distance d_lang stays quite small when the object is at the origin, but increases when the object is pushed. Is there a specific reason why changes in the LLM output for the former (“The object’s x, y position is A,B”) results in smaller language distance than the latter (“The distance is X units.”)? On this note, it would be interesting to see how the language distance changes as a function of the robot state - is it more or less linear?
Since the LLM output is constrained to the “’Description: The cube’s x, y position is ___ .’” and “’Description: the distance is ___ units.’”, I can see how the language distance could serve as a metric between the two behaviors, but would it really correspond to something much different than Euclidean distance on description-level? For example, “The distance is 4 units” vs “The distance is 6 units”.

## Summary:
Overall I think it’s a very well-written paper, with a good mathematical basis and satisfactory results. The results are promising, and I think there are good direction for building upon the methods, for example with VLMs. In my opinion, the proposed method has a lot of potential which is not fully utilized. My main concern is that the examples shown in the paper can be easily represented in a simpler way without an LLM. I can see that there are situations where this might not be the case, and showing these will make a much stronger argument for the proposed approach.

---

> ### Author Response · Authors · 2024-11-21
>
> Before beginning, we sincerely appreciate the reviewer Dsd7 for the detailed, thoughtful review, and bringing the interesting discussion points. We take your review seriously and try to develop our work. Let us give a step-by-step answer.
> \
> \
> **(W1) More complex tasks**
>
> We recognize this as the most significant concern, highlighted by three reviewers. We appreciate your feedback and have extended our method to address more challenging tasks requiring deeper semantic understanding of the scene. Specifically, we evaluated our approach on the task of:
>
> ***“Manipulating Edible Objects”***
>
> For each episode:
> - We randomly sample one edible and one non-edible object from a set of two edible and two non-edible objects:
>   - Edible objects: banana, meat_can
>   - Non-edible objects: foam_brick, mug
> - We ask the LLM for a state description (e.g., position) of the edible objects.
> - LGSD then seeks diverse states for the edible objects only.
>
> \
> We evaluated the trained model with randomly sampled skills across various configurations of objects and initial states. LGSD successfully manipulated the edible objects according to arbitrary configurations. A **video** demonstrating these learned skills is included **in the supplementary materials**.
> \
> \
> **(W2) Video request**
>
> We have included a video containing the following:
> - Demonstrations of the newly added complex task, “Manipulating Edible Objects”.
> - Constraining skill-space experiments for both Ant and Franka environments.
> - Zero-shot natural language following agents trained with LGSD for both Ant and Franka environments.
>
> Please refer to the attached video in the supplementary materials.

---

> ### Author Response · Authors · 2024-11-21
>
> **(Q1) Dimensionality of the skill-space**
>
> We used a skill dimension of 3 for the Franka manipulation task and 2 for the Ant locomotion task. To ensure reproducibility, we have provided all hyperparameters used, including skill dimensions, in Table 3 of Appendix F.2.2.
>
> After reading your review, we conducted ablation experiments with varying skill dimensions [3,6,9,12,18] for the Franka manipulation task. We measured average moved distance of the object as the performance metric. For each skill dimension, we ran four experiments with different random seeds to ensure robustness.
>
> | Steps        | 5k          | 10k         | 15k         | 20k         |
> |--------------|-------------------|-------------------|-------------------|-------------------|
> | **Skill Dim: 3**  | 10.36 ± 11.08     | 28.05 ± 9.29      | 41.88 ± 5.71      | 49.07 ± 1.00      |
> | **Skill Dim: 6**  | **17.05 ± 9.67**      | 27.58 ± 15.43     | 32.92 ± 14.55     | 37.00 ± 11.87     |
> | **Skill Dim: 9**  | 6.68 ± 4.15       | 32.00 ± 4.84      | **47.98 ± 2.11**      | **49.42 ± 0.78**      |
> | **Skill Dim: 12** | 5.03 ± 8.07       | 25.02 ± 14.23     | 32.45 ± 13.73     | 39.98 ± 6.73      |
> | **Skill Dim: 18** | 8.10 ± 4.52       | **32.10 ± 10.25**     | 38.78 ± 12.77     | 40.30 ± 13.05     |
>
> As shown in the table, we observed that the performances were not very sensitive to skill dimensions. We believe this is because, as long as the skill dimension is sufficiently high to encompass the state manifold (as determined by the distance function), the performance remains stable. However, if the degrees of freedom (DoF) of the system increase—such as in a humanoid performing diverse actions—and the distance metric captures all subtle differences between those actions, the state manifold's dimensionality would increase. In such cases, a higher skill dimension may be necessary for the latent space to adequately represent the manifold.
> \
> \
> **(Q2) Explanation for the performance of the baselines**
>
> Yes, our task differs from the one used in METRA in two key aspects:
>
> 1. *Termination Condition*
>
>     The first difference lies in the termination condition. In our setup, an episode is instantly terminated if the agent’s torso height drops below a threshold of 0.31 m. This threshold is the default value used in the Ant environment from the official IsaacGymEnvs library. We believe this condition encourages the agent to walk rather than "roll," which can occur when attempting to maximize all possible state dimensions without constraints, as seen in METRA, LSD, and CSD. Since LGSD leverages the LLM to focus on dimensions that produce distinct language descriptions, this termination condition did not affect LGSD's performance.
>
> 2. *Observation Space*
>
>     Another difference is that we use a higher-dimensional observation space, as detailed in Appendix F.2.1. Our observation space has 39 dimensions, compared to the 26 dimensions in the original Metra task. One notable addition is the inclusion of the previous step's action in the observation space. This modification may disadvantage baseline algorithms, as maximizing state differences for consecutive states in these additional dimensions often results in jiggly and unstable motions. We have also noted this difference in Appendix F.1.
> \
> \
> **(Q3) More complex task**
>
> Yes, we agree that a simple logic gate with a manually designed distance function based on object and end-effector's position could learn a similar set of skills. However, prompting with natural language provides a more convenient and scalable way to induce behaviors compared to manually designing distance functions for each behavior set. Taking this feedback into account, we moved forward and expanded our approach.
>
> To highlight the true value of leveraging an LLM’s *“common knowledge”*, we tested our framework on more advanced tasks that explicitly depend on this capability. For further details, please refer to the attached video.

---

> ### Author Response · Authors · 2024-11-21
>
> **(Q4) LSD with oracle priors**
>
> Your guess was correct. We conducted two experiments with different initial settings:
> - **“init_far”**: The initial distance between the robot end-effector and the object is 1.0 m (Fig. 5).
> - **“init_close”**: The initial distance between the robot end-effector and the object is 0.155 m.
>
> Using these two setups, we trained LSD with oracle observations provided to the encoder $\phi(s)$. Here, oracle observation refers to using only the (x,y,z) position of the object. Although we trained with just one seed, we tested the model using 100 randomly sampled skills. We then measured the average distance the object was moved from the origin.
>
>
>
> |        | Average Moved Distance   |
> |------------------------|--------------------------|
> | **init_far**               | 0.0 ± 0.0               |
> | **init_close**             | 49.15 ± 0.62            |
>
>
> In the “init_far” setting, the agent struggles to make contact with the cube, resulting in a moved distance of 0. Without a learning signal, the robot arm aimlessly moves through the air. However, when the initial condition is changed to “init_close”, exploration is no longer a significant challenge, and the agent quickly learns how to manipulate objects within its operational boundary.
> \
> \
> **(Q5) language distance**
>
> Thank you for raising such an interesting question. We appreciate your thoughts, as they gave us an opportunity to reflect on why this behavior occurs. Based on our analysis, we conclude that the issue is not about how language distances are distributed between different descriptions, but rather about **“path sharing.”**
>
> If you closely examine Figure 12, particularly the blue curve in the real world from the 0th to the 90th step, you’ll notice that the trajectories share nearly identical paths when approaching the cube, even though their sampled skills are aimed in different directions.
>
>  For training representation function $\phi:s \rightarrow z $,  we maximize following objective , $( \phi(s’) - \phi(s) )^T * z$ , under language distance constraints. If two different episodes share same trajectory in state space, they will share same trajectory in the latent space too. And you want the vector $\phi(s’) - \phi(s)$ to be aligned with all different $z$. In this setting, we have no incentives for increasing $||\phi(s’) - \phi(s)||$, because being dot producted with all different directions of $z$ will cancel out the effect of increases, hence has no effect on the objective. Increasing the latent distance in one direction would increase the objective function for some z values, but simultaneously decrease it for other halfs.
>
> In other words, when paths in the state space are shared, the algorithm cannot separate off the $\phi(s)$ and $\phi(s')$ in the latent space. This explains why the latent distance remains small until the object remains still. Once the robot arm begins pushing the object in diverse directions, the trajectories no longer overlap, and the objective function can safely increase the $||\phi(s’) - \phi(s)||$ toward the direction of the sampled $z$.
>
> A follow-up question might be: why do different skills share the initial path toward the object? While we are not entirely certain, it could be due to suboptimal optimization. Once the first path that touches the object is discovered, excessive optimization might cause the agent to quickly converge to a similar initial path, and then focusing more on diversifying the object's movement rather than varying the approach paths.

---

> > ### Comment · Reviewer_Dsd7 · 2024-11-26
> > **Response to Authors' rebuttal**
> >
> > First, I'd like to thank the Authors for their significant efforts in addressing all of the concerns raised in the review, and the overall high quality of their rebuttal. I am happy with the answers and discussions, especially given that they are backed by additional experimental validation. I believe the newly added  "Manipulating edible objects" scenario, while still relatively simple, does better showcase the strengths of using an LLM to guide the skills. As such, I'm happy to re-evaluate my initial score and recommend accepting the paper.

---

> ### Author Response · Authors · 2024-11-25
>
> Dear Reviewer Dsd7, thank you for your efforts to reviewing our submission. As we are getting closer to the end of the discussion period (ending tomorrow) , we wanted to kindly remind you. We have made every effort to address the concerns raised and would greatly appreciate your feedback on whether you had the opportunity to review our responses and if they addressed your points.

---

### Official Review · Reviewer_XxQY · 2024-11-04

**Soundness:** 3
**Presentation:** 3
**Contribution:** 3
**Rating:** 6
**Confidence:** 4

**Summary:**

Rather than to maximize state coverage, this work introduces LGSD to constrain skill discovery inside a semantically meaningful subspace, created by LLMs with crafted user prompts.

**Strengths:**

- The method tried to link skill states with their semantic meaning, resulting in meaningful and language-controllable skills.
- LGSD provides a theoretical foundation for the language-distance metric as a pseudometric, detailing how it can approximate semantic diversity.

**Weaknesses:**

- The experimental scenarios are simple, in which the exampled prompts and semantically controlled spaces are easy to follow yet fail to demonstrate the generalizablity and scalability --- after all, the method relies much on the description of states. LGSD’s dependence on LLMs for real-time distance evaluation might limit scalability to complex, real-time environments.
- As I understand, users have to provide specified "skill constraints"  (via prompts, such as "move north" etc.), then how can it still be called "skill discovery" since users are specifying skills?
- The comparison with some baselines is somehow unfair since they lack the prior knowledge of users or any language embedding computation. A better comparison should be considered.

**Questions:**

- Is it correct that for each state $s_t$, the LLM will be called to generate a description in order to compute $d(s_{t-1}, s_t)$? If so, how fast is the training?
- How will LGSD generalize to complex scenarios? Is it necessary for experts to compose (skill)prompts for each possible skill?

---

> ### Author Response · Authors · 2024-11-21
>
> Thank you for your review. We have organized our responses by topic to address your concerns and questions in detail.
> \
> \
> **(W1.1) More complex tasks**
>
> We recognize this as the most significant concern, highlighted by three reviewers. We appreciate your feedback and have extended our method to address more challenging tasks requiring deeper semantic understanding of the scene. Specifically, we evaluated our approach on the task of:
>
> ***“Manipulating Edible Objects”***
>
> For each episode:
> - We randomly sample one edible and one non-edible object from a set of two edible and two non-edible objects:
>   - Edible objects: banana, meat_can
>   - Non-edible objects: foam_brick, mug
> - We ask the LLM for a state description (e.g., position) of the edible objects.
> - LGSD then seeks diverse states for the edible objects only.
>
> \
> We evaluated the trained model with randomly sampled skills across various configurations of objects and initial states. LGSD successfully manipulated the edible objects according to arbitrary configurations. A **video** demonstrating these learned skills is included **in the supplementary materials**.
> \
> \
> **(W1.2) Dependence on real-time LLM evaluation, scalability concerns**
>
> We would like to clarify that our method does not rely on real-time LLM inference. The LLM is only queried during training. Once training is complete, we simply use a resulting policy network, which is identical to those in other skill discovery methods, ensuring scalability and efficiency during deployment.
> \
> \
> **(W2) Reconciliation between constraining skills and discovering skills**
>
> The ultimate goal of our work is to discover a set of “meaningful” skills. But what constitutes meaningfulness? This depends on the potential downstream use cases for the learned skills. The *constraining* aspect of our method plays a crucial role here. Unlike other skill discovery methods, LGSD allows constraints on the resulting behaviors prior to the training.
>
> Consider it analogous to training a good basketball player: you teach basketball-related skills such as dribbling, shooting, and jumping, but not punching or kicking, which might be useful in mixed martial arts.
>
> **In summary, LGSD constrains the skill “space”, not the individual skill.** Within this constrained subspace, any skills can be learned. In our experiments, we hypothesized that a robot arm would be more interested in manipulating objects rather than aimlessly moving in the air. LGSD, through LLM-provided signals, focuses the robot arm on desired skills, leading to object manipulation into diverse directions, while other methods often produce skills involving random arm movements.
> \
> \
> **(W3) Fairness of LGSD vs baselines, in terms of using prior knowledge**
>
> One goal of our work is to integrate the LLM’s prior knowledge into the skill discovery framework and demonstrate its benefits, such as enhanced semantic diversity, sample efficiency, and skill reusability. To this end, we built on the METRA framework and introduced language distance into the algorithm. To validate the impact of this prior knowledge, we compared our approach against state-of-the-art skill discovery methods [1, 2, 3].
>
> For existing LLM-based skill acquisition methods [4, 5], direct comparison was challenging because those works focus on learning single high-fidelity skills, whereas our aim is to discover a diverse set of skills.
>
>
> [1] Seohong Park, Oleh Rybkin, and Sergey Levine. Metra: Scalable unsupervised rl with metric-aware abstraction. In The Twelfth International Conference on Learning Representations, 2023.
>
> [2] Seohong Park, Jongwook Choi, Jaekyeom Kim, Honglak Lee, and Gunhee Kim. Lipschitz-constrained unsupervised skill discovery. In International Conference on Learning Representations, 2021.
>
> [3] Seohong Park, Kimin Lee, Youngwoon Lee, and Pieter Abbeel. Controllability-aware unsupervised skill discovery. In International Conference on Machine Learning, pp. 27225–27245. PMLR, 2023.
>
> [4] Yecheng Jason Ma, William Liang, Guanzhi Wang, De-An Huang, Osbert Bastani, Dinesh Jayaraman, Yuke Zhu, Linxi Fan, and Anima Anandkumar. Eureka: Human-level reward design via coding large language models. In The Twelfth International Conference on Learning Representations, 2023.
>
> [5] Wenhao Yu, Nimrod Gileadi, Chuyuan Fu, Sean Kirmani, Kuang-Huei Lee, Montserrat Gonzalez Arenas, Hao-Tien Lewis Chiang, Tom Erez, Leonard Hasenclever, Jan Humplik, et al. Language to rewards for robotic skill synthesis. In 7th Annual Conference on Robot Learning, 2023.

---

> ### Author Response · Authors · 2024-11-21
>
> **(Q1) Training speed / cost concerns**
>
> Yes, in order to compute $d(s_{t−1},s_t)$, LLM need to be called.
>
> We acknowledge concerns regarding the cost of LLM queries. To address this, we made efforts to minimize the number of LLM calls. Specifically, we cache the outputs of the LLM for states encountered during training. These cached outputs are reused when the agent encounter same states in the future. In addition, we reused them not only within the same training run but also across subsequent experiments. As a result, the LLM is only called when the agent encounters a previously unseen state.
>
> To make this caching mechanism more effective, we discretized the states information for LLM calls. Coarse binning minimizes the number of queries, while fine binning provides denser learning signals, creating a trade-off between query efficiency and learning quality. Depending on the task, the total number of cached queries ranges from a few hundred (e.g., cube_push_north) to tens of thousands (e.g., cube_push_every_direction).
>
> Each training run takes approximately 10-20 hours on an A40 GPU.
> \
> \
> **(Q2) Generalization to more complex scenarios**
>
> While our method enables training a desired set of skills using tailored prompts, specifying prompts for highly complex tasks can be challenging. In such cases, our method serves as a pre-training module, which can then be fine-tuned for complex tasks or used to train a high-level controller over the learned skills. As shown in Section 5.3, LGSD outperforms baselines in training high-level controllers for tasks like AntMultiGoal, demonstrating its generalizability.

---

> ### Author Response · Authors · 2024-11-25
>
> Dear Reviewer XxQY, thank you for your efforts to reviewing our submission. As we are getting closer to the end of the discussion period (ending tomorrow) , we wanted to kindly remind you. We have made every effort to address the concerns raised and would greatly appreciate your feedback on whether you had the opportunity to review our responses and if they addressed your points.

---

> ### Author Response · Authors · 2024-12-01
>
> Dear Reviewer XxQY,
>
> There is only one day remaining, and we sincerely hope to hear from you regarding whether you had the chance to review our responses and if they addressed your concerns. We also put significant effort into creating the video, so we kindly ask that you take a look and share your thoughts. Thank you for your time and consideration.

---

### Official Review · Reviewer_UwKy · 2024-11-04

**Soundness:** 3
**Presentation:** 3
**Contribution:** 3
**Rating:** 6
**Confidence:** 3

**Summary:**

This paper proposed to incorporate semantic diversity in skill discovery.  The authors show that the proposed approach with language guidance from LLMs outperforms existing skill discovery baselines. The proposed approach is demonstrated in both locomotion and manipulation tasks.

**Strengths:**

- A theoretical proof of language-distance as a valid pseudometric is provided.
- Both locomotion and manipulation tasks have been demonstrated, showing the effectiveness of the proposed approach.
- The paper is well presented.

**Weaknesses:**

- More complex scenarios should be designed to make the use of LLMs necessary rather than simple positions.
- The lack of real-world experiments makes the proposed approach less convincing.
- A video showing the robot in action should be included to give readers an intuitive understanding of learning performance.

**Questions:**

- In Fig. 4, the coverage for the manipulation task is worse than the locomotion task. More analysis or explanations would be necessary to understand the limitations of the proposed approach.
- In Fig. 7, showing the coverage for LGSD in the whole space would be better. Although the proposed approach may have a larger coverage than the baselines, it is less evenly distributed compared to CSD. Is this always the case for different training runs? Any analysis or explanations would be helpful.

---

> ### Author Response · Authors · 2024-11-21
>
> We appreciate your review. Below, we provide a topic-wise response to address your concerns and questions.
> \
> \
> **(W1) More complex tasks**
>
> We recognize this as the most significant concern, highlighted by three reviewers. We appreciate your feedback and have extended our method to address more challenging tasks requiring deeper semantic understanding of the scene. Specifically, we evaluated our approach on the task of:
>
> ***“Manipulating Edible Objects”***
>
> For each episode:
> - We randomly sample one edible and one non-edible object from a set of two edible and two non-edible objects:
>   - Edible objects: banana, meat_can
>   - Non-edible objects: foam_brick, mug
> - We ask the LLM for a state description (e.g., position) of the edible objects.
> - LGSD then seeks diverse states for the edible objects only.
>
> \
> We evaluated the trained model with randomly sampled skills across various configurations of objects and initial states. LGSD successfully manipulated the edible objects according to arbitrary configurations. A **video** demonstrating these learned skills is included **in the supplementary materials**.
> \
> \
> **(W2) Real-world experiments**
>
> We agree that including real-world experiments would strengthen our work. However, we kindly ask the reviewer to consider that several existing state-of-the-art works on skill discovery [1, 2, 3] present their results exclusively in simulation. While real-world applications would be a valuable addition, we see this as an exciting direction for future research.
>
> [1] Seohong Park, Oleh Rybkin, and Sergey Levine. Metra: Scalable unsupervised rl with metric-aware abstraction. In The Twelfth International Conference on Learning Representations, 2023.
>
> [2] Seohong Park, Jongwook Choi, Jaekyeom Kim, Honglak Lee, and Gunhee Kim. Lipschitz-constrained unsupervised skill discovery. In International Conference on Learning Representations, 2021.
>
> [3] Seohong Park, Kimin Lee, Youngwoon Lee, and Pieter Abbeel. Controllability-aware unsupervised skill discovery. In International Conference on Machine Learning, pp. 27225–27245. PMLR, 2023.
> \
> \
> **(W3) Video request**
>
> We have included a video containing the following:
> - Demonstrations of the newly added complex task, “Manipulating Edible Objects”.
> - Constraining skill-space experiments for both Ant and Franka environments.
> - Zero-shot natural language following agents trained with LGSD for both Ant and Franka environments.
>
> Please refer to the attached video in the supplementary materials.

---

> ### Author Response · Authors · 2024-11-21
>
> **(Q1) Coverage comparison between manipulation vs locomotion**
>
> In the locomotion task, starting from the origin, every path taken in the early stages of training is rewarded, as the reward is based on any change in the agent’s position. This naturally leads to more dispersed trajectories and higher coverage. In contrast, for the manipulation task in Section 5.1, the agent only receives a reward after the robot arm makes contact with one of the cube’s four sides and pushes it. When this occurs, the reinforcement learning (RL) algorithm increases the probability of repeating that specific trajectory, while reducing the likelihood of others.
>
> In other words, pushing all sides of the object is less likely to happen unless the robot arm makes contact with all four sides of the cube during a similar training stage. This results in trajectories that predominantly focus on pushing a specific side of the object, leading to less dispersed coverage. By contrast, the locomotion task incentivizes movement in all directions equally.
>
>
> \
> **(Q2) Analysis of LGSD vs CSD in Fig.7**
>
> When the robot arm makes initial contact with one side of the cube, it largely determines the possible future trajectories. For example, if the robot starts pushing the northern side of the cube, it cannot produce a trajectory of moving the cube northward. The most natural subsequent movements would be pushing the cube southward, or at most, slightly southeast or southwest.
>
> This highlights a key difference between LGSD and other methods, including CSD. In CSD training, each skill tends to push only one side of the cube, resulting in the cube being moved predominantly in the same direction as presented in Fig. 7, In contrast, LGSD is capable of pushing different sides of the cube depending on the sampled skill.
>
> How is this achieved? LGSD provides rewards not only for pushing the cube but also for reaching the cube. Using the prompts suggested in Fig. 3, LGSD rewards the intermediate paths taken by the end-effector as it moves from its initial position to the cube. This encourages exploration of diverse trajectories, ultimately enabling the agent to push different sides of the cube.
>
> On the other hand, CSD incentivizes state transitions that are less likely to occur. As a result, when the robot arm manages to push the cube in a specific trajectory, the learning algorithm heavily focuses on that specific trajectory. This makes it unlikely to learn a diverse set of skills capable of pushing both the northern and southern sides of the cube.

---

> ### Author Response · Authors · 2024-11-25
>
> Dear Reviewer UwKy, thank you for your efforts to reviewing our submission. As we are getting closer to the end of the discussion period (ending tomorrow) , we wanted to kindly remind you. We have made every effort to address the concerns raised and would greatly appreciate your feedback on whether you had the opportunity to review our responses and if they addressed your points.

---

> > ### Comment · Reviewer_UwKy · 2024-11-26
> >
> > Thank you for addressing my concerns through the additional experiments and clarifications. Most of my concerns have been resolved. While demonstrating more complex tasks could further strengthen the contributions, I am pleased with the improvements and happy to increase my initial score.

---

### Meta-Review · Area_Chair_t6Wm · 2024-12-24

**Metareview:**

The paper is well-presented. It proposes a novel method Language Guided Skill Discovery (LGSD), using language distances as for skill learning with LLMs. It demonstrates zero-shot goal tracking and skill discovery in diverse tasks.

Strengths include a mathematically sound approach, effective language-constrained skill prediction, and promising results in new tasks. It includes theoretical validation and simulated demonstrations in locomotion and manipulation tasks.

Weaknesses include limited evidence for the necessity of LLMs, and a lack of real-world experiments. Comparison with baselines is not completely fair since those methods did not use prior language-based knowledge.

**Additional Comments On Reviewer Discussion:**

The reviewers raised concerns about the limited tasks requiring deep semantic understanding to validate generalizability and scalability, missing real-world experiments, and video demonstrations, as well as baseline discrepancies. The authors added a new task requiring deeper semantic understanding and video demonstrations. The authors also added clarifications for evaluations and setup differences.

---

### Decision · Program_Chairs · 2025-01-22

Accept (Poster)